# Chitosans for Tissue Repair and Organ Three-Dimensional (3D) Bioprinting

**DOI:** 10.3390/mi10110765

**Published:** 2019-11-11

**Authors:** Shenglong Li, Xiaohong Tian, Jun Fan, Hao Tong, Qiang Ao, Xiaohong Wang

**Affiliations:** 1Center of 3D Printing & Organ Manufacturing, School of Fundamental Sciences, China Medical University (CMU), No. 77 Puhe Road, Shenyang North New Area, Shenyang 110122, China; lishenglong@cancerhosp-ln-cmu.com (S.L.); xhtian@cmu.edu.cn (X.T.); jfan@cmu.edu.cn (J.F.); tongh007@hotmail.com (H.T.); aoqiang00@163.com (Q.A.); 2Center of Organ Manufacturing, Department of Mechanical Engineering, Tsinghua University, Beijing 100084, China

**Keywords:** 3D bioprinting, chitosan, chitin, tissue repair, rapid prototyping (RP), implantable bioartificial organs

## Abstract

Chitosan is a unique natural resourced polysaccharide derived from chitin with special biocompatibility, biodegradability, and antimicrobial activity. During the past three decades, chitosan has gradually become an excellent candidate for various biomedical applications with prominent characteristics. Chitosan molecules can be chemically modified, adapting to all kinds of cells in the body, and endowed with specific biochemical and physiological functions. In this review, the intrinsic/extrinsic properties of chitosan molecules in skin, bone, cartilage, liver tissue repair, and organ three-dimensional (3D) bioprinting have been outlined. Several successful models for large scale-up vascularized and innervated organ 3D bioprinting have been demonstrated. Challenges and perspectives in future complex organ 3D bioprinting areas have been analyzed.

## 1. Introduction

An organ is a combination of multiple tissues that provide specific physiological functions for human body. There are about 80 organs in the human body, dominating every physiological activity [1]. Clinically, organ failure is the leading cause of mortality all over the world, and it is often closely related to some chronic and acute diseases [2,3]. As a result, one failing organ can break down the whole physiological system of the human body. 

At present, allogeneic organ transplantation is the only cure for the failing internal organs, such as the liver, kidney, and heart. However, it is facing many limitations in clinical applications. First, there is a desperate shortage of donor organs. In another word, the donor organ is seriously short. Take the year of 2013 as an example, there were 117,040 patients in the United States of America who needed organ transplantation, but only 28,053 of them are fortunate to get suitable donors [4]. Currently, there are over 34 million surgical procedures in America involved in the treatment of organ failures per year. Less than one out of ten patients can be saved by organ donations [5]. Second, there are serious side effects of immunosuppressive drugs. The donor organs are allogeneic sources and the patients need to take a life-long immunosuppressive treatment. Since the side effects of immunosuppressive complications are severe, the ability of resisting infectious diseases can be obviously weakened. Third, the surgical cost is very high. Organ transplantation can bring huge economic burdens to the ordinary patients and occupy enormous medical resources [6].

During the last several decades, the severity of donor organ shortage, the life-long treatment of allograft rejection, the side effect of immunosuppressive therapy, and the extremely high cost of allogeneic organ transplantation have activated numerous strategies for tissue repair and organ manufacturing [7,8,9,10,11]. One typical example is tissue engineering. It has undergone several circulations of ups and downs with the three elements, i.e., porous scaffolds, cells, and growth factors. Nevertheless, organ manufacturing is a complex project that requires multi-disciplinary cooperation, involving a large scope of talents of technologies, such as biology, materials, chemistry, physics, mechanics, informatics, computers, and medicine. Designing and building the physical analogues of organs is only the first small step. It is more critical to make the multiple cell types/extracellular matrices (ECMs), hierarchical vascular, neural and/or lymphatic networks functional in a compacted construct [12,13,14].

Three-dimensional (3D) bioprinting has recently emerged as an extension of traditional rapid prototyping (RP), also named as solid freeform fabrication (SFF) and additive manufacturing (AM), technologies, by using bioactive or cellular components to build constructs in an additive or layer-by-layer methodology for encapsulation and culture of cells. These technologies allow for cell culture in controlled spatial environments. These environments can be tuned to simulate the complexity of in vivo cell growth environments with similar ECMs.

Polymers are large molecules or macromolecules, composed of many repeated subunits or small molecules, with molar masses ranging from thousands to millions. In another word, the units composing polymers derive from molecules of relatively low molecular mass [15]. There are two types of polymers: natural and synthetic. Natural polymers are naturally occurring polymers, such as cellulose, polysaccharide, protein, silk, and fibrinogen. Synthetic polymers are manmade polymers through chemically joining many small monomers together into one giant molecule. The list of synthetic polymers, roughly in order of worldwide demand, includes polyethylene, polypropylene, polystyrene, polyvinyl chloride, synthetic rubber, neoprene, nylon, polyacrylonitrile, phenol formaldehyde resin (or Bakelite), polyvinylbutyral (PVB), silicone, and many more. Because of their broad range of properties, both synthetic and natural polymers play essential and ubiquitous roles in everyday life. More than 330 million tons of these polymers are made every year (2015) [16]. Chitosan as a special natural polymer has attracted great attention both in tissue repair and organ 3D bioprinting areas. 

## 2. Three-Dimensional (3D) Bioprinting

### 2.1. The Concept of Organ 3D Bioprinting

3D printing, traditionally termed as RP, SFF, and AM, is a series of material processing technologies based on the dispersion-accumulation principle of computer-aided manufacturing (CAM). Generally, an object can be divided into numerous two-dimensional (2D) layers before 3D printing with a defined thickness. The 2D layers are sequentially piled up by selectively adding the desired materials in a high reproductive additive manner under the instruction of computer-aided design (CAD) models [17,18,19,20]. 

Organ 3D bioprinting is the utilization of advanced 3D printing technologies to assemble multiple cell types, including stem cells/growth factors, along with other biomaterials in a layer-by-layer fashion to produce bioartificial organs that maximally imitate their natural counterparts with respect to anatomical structures, material components, and physiological functions (Figure 1) [12,13,14]. It mainly consists of four aspects, such as cell extraction (i.e., biological part), data collection (i.e., informatical part), starting material preparation (i.e., biomaterial part), and manufacturing (i.e., processing part). Patient-specific organ images, such as magnetic resonance imaging (MRI) and computerized tomography (CT) can be easily transferred into CAD models for customized organ manufacturing with predefined geometrical shapes, material (e.g., cells, growth factors, polymers, ECMs, drugs) components, and physiological functions [21,22,23,24]. Over the last decade, organ 3D bioprinting technologies have made a great contribution to various biomedical fields. 

### 2.2. Polymers as “Bioinks” for 3D Bioprinting

As stated above, polymers are large molecules made up of many small and identical repeating units bonded by covalent bonds. Theoretically, any polymer solutions or hydrogels holding the sol-gel transition property can be printed in layers under the instruction of CAD models [25,26]. Living cells and growth factors can be encapsulated into the polymeric solutions or hydrogels for 3D tissue and organ construction. Actually, only few natural and synthetic polymers and their combinations can be used as “bioinks” for tissue and organ 3D bioprinting at mild temperatures or cell endurable conditions. The layer-by-layer construction processes depend largely on the liquid polymer solution transformation capabilities before, during, and after the 3D bioprinting. There are many different sol-gel transformation forms, such as physical (reversible), chemical (reversible or irreversible), and biochemical (i.e., enzymic) cross-linking. Compared with synthetic polymers, most of the natural polymeric hydrogels can provide cells with suitable environments to survive. Currently, natural polymeric hydrogels are the dominate components of “bioinks” for 3D bioprinting. 

There are three major types of organ 3D bioprinting technologies (Figure 2): (A) multi-nozzle extrusion-based bioprinting (a: pneumatic; b: Piston); (B) multi-nozzle inkjet-based bioprinting (a: heater; b: piezoelectric actuator); (C) multi-channel laser-assisted bioprinting. Cellular behaviors are easily manipulated within the polymeric hydrogels, via adjusting the physical, chemical, biochemical, and physiological properties of the 3D printable polymers. It is surprising that all the bottleneck problems, such as large scale-up tissue/organ manufacturing, living tissue/organ preservation, hierarchical vascular/neural network construction, and partly/fully stem cell engagement, which have perplexed tissue engineers and other researchers for more than several decades, have been overcome by a single scientist, the corresponding author of this article herself, via several series of automatic and semiautomatic layer-by-layer material integration and step-by-step stem cell inducement strategies [12,13,14].

Thus, the purpose of organ 3D bioprinting is to design and manufacture bioartificial organs using polymeric materials, cells, bioactive agents, and advanced 3D printers. Defining and creating appropriate polymeric “bioinks” is a critical step in bioartificial organ 3D bioprinting [12,13,14]. Natural polymers, such as alginate, gelatin, hyaluronate, and chitosan, have been chosen as the preferable candidates for organ 3D bioprinting because of the specific properties, such as biocompatible, bioprintable, biodegradable, bioavailable, and biostable. Particularly, these natural polymers can be easily predesigned as the ECMs of each tissue in a natural organ. 

## 3. Properties of Chitosan as a Natural Polymer

### 3.1. Resource of Chitosan

Chitin is one of the most abundant natural polymers in organism [27]. It is the second most abundant natural polymer next to cellulose, consisting of 2-acetamido-2-deoxy-d-glucose through a (1-4) linkage, extracted from the shells of marine crustaceans, insects, or fungi. Thus, the structural formula of chitin is poly-(1-4)-*N*-acetyl-glucosamine. Because of its facility to generate long-chain polymer structure, chitin is vitally important for some biological structure formation, such as cell walls in fungi and yeast, and exoskeleton of many invertebrates in shrimps and crabs. The extractive product is white, slightly pearly luster, and translucent sheet solid. Because of its insolubility in water and most of the organic solvents, chitin has limited applications in biomedical fields. 

Chitosan, a derivate of chitin, is a linear polysaccharide, or carbohydrate polymer, derived from partial deacetylation of natural chitin. The deacetylation of chitin is conducted by chemical hydrolysis in alkaline conditions, using concentrated alkali water, or enzymatic hydrolysis with chitin deacetylase [28,29,30]. As its origin chitin, chitosan is highly available in nature.

### 3.2. Physico-Chemical Properties of Chitosan

Chitosan is a linear carbohydrate polymer derived from chitin with a structural similarity to glycosaminoglycan, a component of ECMs (Figure 3) [31,32,33,34,35,36,37,38]. Its chemical name is (1,4) -2-amino-2-deoxy-beta-*D*-glucan, a copolymer of randomly located (1-4)-2-amino-2-deoxy-d-glucan (d-glucosamine) and (1-4)-2-acetamido-2-deoxy-d-glucan (*N*-acetyl d-glucosamine) units. The number of amino groups as a ratio between d-glucosamine to the sum of d-glucosamine and *N*-acetyl d-glucosamine is indicated as a deacetylation degree (DD) and ordinarily should be larger than 60%. Some chitosans (e.g., 50% deacetylation degree) are water-soluble, while most of the chitosans are acid-soluble. Thus, chitosan can be recognized as a semi-natural positively charged polysaccharide at acidic conditions. Chitosan molecules can be modified through changing the functional groups, such as OH, and NH_2_ by -COCH_3_, -CH_3_, -CH_2_COOH, -SO_3_H, -PO(OH)_2_, etc. 

### 3.3. Prominent Characteristics of Chitosan for 3D Bioprinting

The biophysical characters of chitosan relay on a few factors, such as the molecular weight, DD, and purity of the molecules. Compared with animal derived natural polymers, such as collagen, gelatin, and fibrin, the degradation rate of chitosan is relatively slow. The biophysical properties of chitosan can be easily adjusted through changing the deacetylation rate of the original chitin and the molecular weight of the product. 

Chitosan molecules possess several important properties, such as biocompatibility, biodegradability, antibacterial activity, non-antigenicity, and bioadsorbility. The solubility of chitosan in diluted acids is pH dependent through protonation of amino groups of the d-glucosamine residues. The availability of protonated amino groups enables chitosan to form complexes with metal ions, natural or synthetic anionic poly(acrylic acid) polymers, lipids, proteins, and deoxyribonucleic acid (DNA) [39,40,41]. The cationic feature of chitosan molecules favors the formation of gel particles through electrostatic interactions, with, e.g., sodium sulfate employed as a precipitant [39,40,41]. 

Stress should be given to the cationic feature of chitosan molecules. Few polymers in nature possess such special characters. The positive charged chitosan molecules can interact with hydrophobic components, giving rise to amphiphilic particles with great self-assembly and encapsulation capabilities. The polycationic nature of chitosan at a mild acidic condition allows the immobilization of negatively charged enzymes, proteins, and DNA for gene delivery and other composite biomaterial formation [42,43]. It has proven that appropriate interactions between chitosan molecules and drugs can produce expected pharmacological effect at the target site. The hydrophilic structure of chitosan promotes almost all cell types to adhere and proliferate on its scaffolds and makes chitosan a good candidate for tissue repair and organ 3D bioprinting. 

## 4. Chitosan-Base Polymers in Tissue Repair and 3D Bioprinting

### 4.1. Antimicrobial Activities for Skin Regeneration

Chitosan and its derivations have shown a lot of distinctive characters in skin regeneration with antimicrobial activities. The mechanisms of antibiosis for Gram positive and Gram-negative bacteria are different. The differences are closely related to the composition of the bacterial walls [44,45]. It is supposed that chitosan presents antimicrobial activities through effectively interacting with the outer membrane or cytoderm of the bacteria, probably, electrostatic attraction and osmotic pressure taken upon the dominant roles in the interactions.

In Gram-positive bacteria, such as staphylococcus and streptococcus, the cytoderm consists of peptidoglycan with a thickness of 20–80 nm. *N*-acetylmuramic acids in the cytoderm can absorb negatively charged teichoic acids through covalent linkages. At the same time, lipopolyteichoic acids form covalent bonds with the cytoplasmic membrane. When teichoic acids form a layer of high-density charges in the cytoderm, the cytoderm is strengthened and the ion transportation through the outer surface layers can be restrained. Compared with Gram-positive bacteria, the peptidoglycan stratum in the cytoderm of Gram-negative bacteria is relatively thin, which lies over the cytoplasmic membrane. This complexity is further covered by an additional outer envelope membrane, with the fundamental elements of lipoprotein and lipopolysaccharide. 

In acidic condition (especially, pH ˂6), the quaternary ammonium groups (R-NH3+) in chitosan molecules can competitively integrate with the divalent metal ions, such as Ca^2+^ and Mg^2+^. To maintain the balance of voltage, polyanions often passively bind with the cytoderm leave. This binding can lead to the cytoderm to be hydrolyzed and cause the leakage of intracellular components [44,45]. The hydrolysis of peptidoglycans also results in an increased electrical interaction, leading to the enhancement of solute conductivity and cytoplasmic β-galactosidase release in the cell suspensions [46,47,48,49]. 

Thus, the reason of chitosan affecting the permeability of the outer cytoderm is through forming an ionic type of bonding and preventing the intracellular transport of nutrients into the cells. This type of tunnel also increases the internal osmotic pressure, leading to the apoptosis of cells because of the lack of nutrients [50]. Some other studies have shown that chitosan can penetrate the multilayered (murein cross-linked) bacterial walls as well as the cytoplasmic membrane. Chitosan destroys the bacterial cells by binding to the DNA which prevents DNA transcription and interrupts protein and m-ribonucleic acid (mRNA) synthesis [50]. The destructiveness is highly dependent on the capability of chitosan to penetrate the multilayered cell wall and the cytoplasmic membrane. Nevertheless, chitosan has generally been regarded as a membrane disruptor rather than a penetrator during the antimicrobial activities.

Skin regeneration is a complex procedure that contains four dynamic phases—hemostasis, inflammation, proliferation, and tissue remodeling [51]. This dynamic process involves vascular stimulators, ECM components, soluble factors, and various cells. The treatment of skin injuries needs to ensure a very high degree of protection, strong anti-inflammatory effect, and minimum scar formation. 

Chitosan solutions can be made into porous scaffolds with intrinsic antimicrobial properties. The antimicrobial effect of the porous scaffolds can be further promoted through adding other antimicrobial agents for wound healing. In one study, a chitosan-cordycepin hydrogel was prepared via adsorbing negatively charged cordycepin onto the positively charged chitosan molecules without adding any cross-linking agent [52]. In another study, polyethylene terephthalate was 3D printed with a chitosan solution [53]. Each layer of the textile polyethylene terephthalate-chitosan was loaded with chlorhexidine. The stability of the 3D printed porous scaffold was enhanced by a heating system, which also extended the delivery time of chlorhexidine up to seven weeks. Chitosan with other natural polymers can be coprinted into asymmetric membranes, and normally, the lower layer can directly contact the damaged skin [54]. The 3D printed membranes presented efficient antimicrobial capability against methicillin resistant staphylococcus aureus (MRSA) strains during the skin repair processes using a mouse model. The skin repair effect is similar to the commercially available products [55].

In addition to antimicrobial function, chitosan participates in all phases of skin regeneration in many ways. Chitosan molecules can effectively promote the migration of neutrophils, increasing the secretion of IL-8, a potent neutrophil chemokine [56]. This reaction is in correlation with the level of *N*-acetylation [57]. Moreover, chitosan can affect the expression of growth factors by increasing transforming growth factor-1 (TGF-1) expression in early post-injury phase and decreasing it in later stages through binding themselves to anionic growth factors [58,59]. Especially, chitosan molecules with high DD stimulate the proliferation of dermal fibroblasts, allowing fibrous tissue formation and re-epithelialization [60,61]. These unique properties of chitosan molecules make them favorable candidates in skin regeneration.

### 4.2. Hemostatic Activity for Wound Healing

Some chitosan molecules with specific molecular weight and DD demonstrate powerful hemostatic capability, which is independent of the coagulation pathway of the host [62,63,64]. The amine groups in the chitosan molecules can interact directly with coagulation factors, promoting the initiation of coagulation. When the DD of chitosan is 68.36%, chitosan molecules in a solution tend to form mesh-like structures and act with vascular components directly. Whereas higher DD results in stronger hydrogen bonds and crystalline structures within chitosan chains that have limited interaction with red blood cells [65]. The interactions of chitosan molecules with polyelectrolytes can be enhanced when the molecular weight of chitosan is increased, so as to the procoagulation processes [66,67]. There are several chitosan-containing hemostatic products, such as Celox^®^, HemCon^®^, Axiostat^®^, Chitoflex^®^, and Chitoseal^®^, available in the market, which have been approved by the Food and Drug Administration of the United States (FDA) [68].

It is found that 3D bioprinted chitosan/collagen films are useful in wound healing. The host tissues had anaphylaxis reactions to allogenic source collagens. It is necessary to prepare more biocompatible chitosan/collagen substitutes for wound healing in the future. Human keratin-chitosan membrane produced through UV-crosslinking has shown the potential as wound dressing with improved mechanical properties [69]. Chitosan-chondroitin sulfate-based polyelectrolyte complex has shown strong hemostatic capability beside antimicrobial effect for wound healing applications [70]. 3D printed chitosan with positive charged bioactive agents, such as growth factors and cytokines, can promote the wound healing capability. In an attempt, nanoparticles of chitosan generated by ionotropic gelation with tripolyphosphate were loaded with granulocyte-macrophage colony-stimulating factor (GM-CSF). The complexity was freeze-dried afterward, leading to the production of nanocrystalline cellulose–hyaluronic acid combination [71,72]. Polycaprolactone nanofibers loaded with chitosan NPs containing GM-CSF accelerated wound closure phenomenon [73]. 3D printing of chitosan combined with peptides presents the ability of wound closure as well. Bioprinting of cell-laden chitosan hydrogels, containing Ser-Ile-Lys-Val-Ala-Val-chitosan macromers can effectively induce various types of collagen expression, prompting angiogenesis with markers of TGF-1 [74]. Meanwhile, the inflammatory factors, which are not conducive for wound healing, such as TNF-α, IL-1β, and IL-6 mRNA in a mouse skin wound model were significantly inhibited [75]. In some other studies, chitosans were used to enhance the affinity of growth factors. A 3D printed chitosan scaffold containing heparin-like polysaccharide (2-*N*, 6-*O*-sulfated) demonstrated an enhanced capability to attract vascular endothelial cells and induce the secretion of growth factors because of the high sulfonation degree [76].

### 4.3. Three-Dimensional Constructs for Bone Rehabilitation

Traditionally, chitosan membrane is one of the commonly used biomaterials in biomedical and clinical applications. It can be prepared through various technologies, such as electrospinning, thermal induced phase separation and self-assembly. During electrospinning, chitosan fibers are deposited irregularly to form non-woven fibrous membranes. The physical structures of the non-woven fibrous membranes are similar to those of natural ECMs. The shortages of the non-woven fibrous membranes to be used as tissue engineering scaffolds are the small pore sizes and weak mechanical strengths. On the one side, the pore sizes of the fibrous membranes are too small to let cells grow in [77]. For example, Sajesh et al. prepared a chitosan fibrous membrane through electrospinning [78]. The tensile strength of the fibrous membrane was 10 MPa, the average pore size was 5 um, and the porosity was over 80%. Shalumon found that in a high chitosan-contenting membrane, the tensile strength was only 1.5 MPa, which was lower than that of a commonly clinically used bone regeneration membrane [79]. The tensile strength of chitosan fibrous membrane needs to be improved. The pore size of chitosan membranes prepared through chitosan molecule self-assembly is also small, and the diameter of the pores is easily affected by the concentration of the chitosan molecules, solution pH, temperature, and other factors. Some scholars used sodium chloride as pore-forming agent to obtain large pores in the chitosan membranes via thermal induced phase separation.

Current research shows that calcium phosphate, carbon nanotubes, and hydroxyapatite can increase the mechanical properties of the chitosan scaffolds to some degree [80]. For example, Matinfar et al. mixed chitosan and carboxymethyl cellulose (CMC), and reinforced with whisker-like biphasic and triphasic calcium phosphate fibers as bone repair scaffolds [81]. The composite chitosan/CMC were obtained by freeze drying. The composite scaffolds exhibited desirable microstructures with high porosity (61–75%) and interconnected pores in range of 35–200 μm. Addition of CMC to chitosan solution led to a significant improvement in the mechanical properties (up to 150%) but did not affect the water uptake ability and biocompatibility. The composite chitosan/CMC scaffolds reinforced with 50 wt% triphasic fibers were superior in terms of mechanical and biological properties and showed compressive strength and modulus of 150 kPa and 3.08 MPa, respectively, which is up to 300% greater than pure chitosan scaffolds. Bi et al. prepared a chitosan-containing composite scaffold and seeded with osteoblasts. It was found that a large number of osteoblasts adhered on the scaffold and proliferated inside the go-through pores [82]. When the chitosan-containing composite scaffold was implanted into rats with skull-parietal bone loss, new bone formed at the edge of the bone loss site and the center of the scaffold in 2 weeks. After five weeks’ implantation, new bone mass was significantly higher than that of the blank control.

With the introduction of 3D printing technologies in tissue engineering, the physical, biochemical, and physiological properties of the 3D printed chitosan scaffolds can be greatly improved. In a 3D printed chitosan scaffolds, osteoblasts grew along the computer controlled go-through channels and formed trabecula structures. Meanwhile blood vessels are easy to form along the go-through channels with the addition of endothelial cells [83]. Emphasis should be given to those growth factors, i.e., polypeptides, that can bind to specific cell membrane receptors to control cell destiny and regulate cell functions. Osteoinductive growth factors include vascular endothelial factor (VEGF), bone morphogenetic protein (BMP), platelet-derived growth factor (PDGF), and TGF, etc. Kjalarsdttir et al. cultured mouse fibroblasts with BMP-2 encapsulated in chitosan microsphere through a ion cross-linking method. The results showed that the encapsulation rate of chitosan microspheres was over 80% with a slow growth factor releasing rate. The sustainable releasing time attained 30 days. When the rhBMP-2 adsorbed chitosan microspheres were compounded on collagen sponge scaffolds and implanted into rabbits with radial segmental defects, the rhBMP-2-adsorbed chitosan microsphere scaffolds had more new bone mass than that of the control rhBMP-2/collagen scaffolds 12 weeks after the implantation, indicating that chitosan microspheres as carriers could effectively maintain the biological activity of rhBMP-2 [84]. The chitosan molecules can inhibit the secretion of osteoclasts and promote the proliferation of osteoblasts, thus promoting the bone tissue repair effect. When the VEGF-containing chitosan microspheres were implanted into rat peritoneal adipose tissues, two weeks later, the number of endothelial cells and erythrocytes in the rats was significantly higher than that of the controls. These results suggest that the chitosan-containing 3D scaffolds together with growth factors can effectively promote large bone repair rate [85]. 

In some other researches, the positive charge amino groups in the chitosan molecules can combine with negative charged DNA molecules to form nanoparticles through polyelectrolyte actions. Foreign genes can be transfected into the cells of the body and play some roles in the cell behaviors. For example, Zeng et al. prepared nanoparticles via the reaction of mercaptan-organized chitosan and recombinant plasmid polyelectrolyte. When the recombinant plasmids containing *BMP-4* and *VEGFR1* genes were implanted into rabbits with radius defect, the experimental group had faster bone defect repair speed and more new bone mass compared with the controls [86]. Similarly, chitosan/polyacrylic acid nanofibers had been used as effective carriers of DNA plasmids. These researches have elaborated the active roles of chitosan molecules in bone tissue rehabilitation processes at molecular and cellular levels. 

For large bone repair, a series of pioneering work have been done by the corresponding author of this article herself before 2000 through chemical modification of chitosans (Figure 4, Figure 5 and Figure 6) [31,32,33,34,35,36]. For example, several large bone repair materials have been created by adding phosphorylated chitin (P-chitin), phosphorylated chitosan (P-chitosan), and disodium (1→4)-2-deoxy-2-sulfoamino-β-*D*-glucopyranuronan (S-chitosan) as the additives of biodegradable calcium phosphate cement (CPC) systems. The large bone repair materials are biocompatible, bioabsorbable, osteoconductive and/or osteoinductive. In vitro and in vivo experiments have shown that the bone repair rates and effects are directly related to the functional groups on the chitosan-based molecules and polymer concentrations in the CPCs. There are many different bone repair manners with these materials: some new trabeculae form directly after body fluid infiltration of the implants (Figure 5) [33]; some new trabeculae form following chondrocytes disappearing around the implants (Figure 6) [32]; some new trabeculae form after fibroblast-like cells being swallowed up [36]. The biodegradation rates of the materials have negative relationships with the P-chitin, P-chiosan, and S-chitosan contents. Most of the low concentration samples degrade in 16 weeks. While the high concentration samples disappear around 22 weeks. The fastest bone repair rates comes from those samples containing low concentrations of P-chitin and P-chitosan. Especially, the P-chitosan contained CPCs possess excellent biocompatibilities which can be transferred to trabeculae straightly after body fluid infiltration without any vise reactions or adverse effects, such as hematoma, inflammation, fibrous encapsulation, tissue necrosis, and excessive growth. The degradation rates of P-chitin and P-chitosan contained samples can be adjusted to match the ingrowth speeds of new trabeculae. A mild foreign-body reaction appears in the high P-chitin content samples during the early implantation stages which do not impair the final bone repair effects. 

Later in 2003, the chitosan-based polymers have been 3D printed into hybrid large bone repair scaffolds with synthetic polymers, such as poly(lactic acid-co-glycolic acid) (PLGA), and mineral salts, such as calcium phosphate (TCP). Some of them have multiple functions, such as promoting osteoblast growth and inhibiting osteoclast activity (Figure 7) [87]. Different CAD models have been utilized to manufacture the hybrid scaffolds. Optimal fabrication parameters have been systematically studied through manipulating the processing materials. Furthermore, the microscopic structures, water absorbability, and mechanical properties of the hybrid scaffolds can be easily adjusted through adding different amount of P-chitin, P-chitosan, and S-chitosan. These hybrid scaffolds have been proven to be promising candidates for large hard tissue and organ manufacturing and restoration with later animal tests.

### 4.4. Cartilage Reconstruction

The treatments of cartilage degeneration and damaging are critical tasks in orthopedics. Articular cartilage injury may occur in sports, osteoarthritis, and many other situations. The common clinical therapeutic treatments include microfracture, mosaicplasty, autologous chondrocyte, and biomaterial implantation [88]. However, the vital limitation for cartilage regeneration is the insufficiency of vascular supply in the new cartilage tissues. To generate a graft with not only the capability to promote cartilage regeneration, but also to improve the avascular conditions, is the predominant goal of cartilage tissue reconstruction [89]. 

As stated above, chitosan is a natural polymer with a configurational similitude to sulfate glycosaminoglycans, and presents a similar microenvironment for the proliferation of chondrocyte, simulating ECM synthesis, and promoting chondrogenesis [90,91,92,93,94,95,96]. Compared with solitary alginate beads, the composite chitosan-alginate beads have shown enhanced chondrogenesis capacity when chondrocytes are embedded in. This can reduce the releasing of inflammatory cytokines, such as IL-6 and IL-8, and stimulate cartilage matrix component, such as hyaluronan and aggrecan, synthesis in vitro [97]. The derivates of chitosan have similar physiological activities and functions. For example, carboxymethyl-chitosan significantly reduced iNOS and IL-10 expression in a dose-dependent manner [98]. The addition of hyaluronic acid-chitosan to a pellet co-culture of the human infrapatellar fat pad (IPFP)-derived mesenchymal stem cells (MSCs) with osteoarthritic chondrocytes could increase chondrogenic differentiation [99]. 

From the molecular level, chitosan interacts with collagen through abundant electrostatic interactions between amino and sulfonate groups [100]. Compared with single component scaffolds, the freeze-dried type II collagen-chitosan composite scaffolds have better stiffness and ideal porous structures which are similar to natural cartilage ECMs [101]. It was reported that type II collagen-chitosan scaffolds combined with PLGA in bilayer forms can improve the mechanical and functional properties of the cartilage regeneration grafts [102]. Chitosan-silk fibroin blends can promote the cartilage regeneration efficiency [103,104]. One study found that the ratio of type II collagen to type I collagen of bovine chondrocytes cultured on 300-nm-diameter chitosan fibers was twice as high as that cultured on spongy scaffolds. It is noteworthy that the new macroporous 3D scaffolds prepared with freeze gel method (i.e., Cryogel) are attractive in biomedical fields [39,40,41]. The chitosan agarose gelatin Cryogel has super large pores (85–100 mm long) and good mechanical properties. The compression modulus of 5% Cryogel is about 44 kPa at 15% deformation [105]. When the chitosan-agarose-gelatin gel was used to repair subchondral cartilage defects in female New Zealand rabbits, there was no hypertrophic marker in the formation of hyaline cartilage around the fourth week after implantation. It is chitosan molecules that induce human bone marrow mesenchymal stem cells to differentiate into chondroid spheres by activating mTOR/S6K.

Until present, the widely used chitosan grafts for cartilage repair include cell-laden hydrogels, injectable solutions, and 3D printed scaffolds. Most of the cell-laden hydrogels contain both cellular (e.g., chondrocytes and undifferentiated progenitor chondrocytes) and bioactive molecular components (e.g., peptides, growth factors, and cytokines). These cell-laden hydrogels have low mechanical strength (E ≈ 200 kPa) [88]. Intra-articular injection of the chitosan containing solutions can avoid the trauma of open surgery without affecting chondrocyte colonization and cartilage differentiation. However, the mechanical strengths of the injectable solutions are also very low to meet the clinical requirements. 

Ideally, the biological properties of cartilage grafts should be capable of maintaining adult cell activities and inducing stem cell differentiation. From this point of view, the 3D printed constructs containing suitable biodegradable polymers and cell types are preferable [90,91,92]. In a previous study, a 3D-bioprinted chitosan scaffold attached with TGF-β and BMP-6 and seeded with human IPFP-MSCs demonstrated effective cell proliferation capacity with modified cartilage repair process [105].

It is expected that the 3D printed cartilage grafts hold the following prominent characteristics: (1) The selected biodegradable polymers are highly similar to natural cartilage ECMs, including the microstructures, physicochemical, and biochemical properties; (2) the incorporated cells are able to provide a temporary template to maintain the normal tissue function and synthesize new ECMs [92]; (3) the interconnecting porous structures are beneficial for cell migration and colonization [90]; (4) the 3D printed cell-laden constructs can be implanted in the articular cavity to cover the wound and promote cartilage repair [93].

### 4.5. Three-Dimensional-Bridge for Nerve Repair

Chitosan molecules have excellent neural biocompatibilities [106,107,108]. In vitro studies have shown that chitosan fibers or membranes can effectively promote nerve cell migration and proliferation. The survival time of hippocampal neurons and Schwann cells on chitosan films can be significantly prolonged. When a chitosan conduit was sutured outside a sciatic nerve gap of 10 mm or 15 mm, the motor and sensory functions of the never damaged rats could be totally recovered [109,110,111,112]. This chitosan conduit could reconstruct the long-distance peripheral nerve defect in diabetic rats and achieve similar recovery effects as autologous nerve transplantation [113,114,115]. Neurotrophic factors such as nerve growth factors could mix to the chitosan conduit through 3D bioprinting technology to promote peripheral nerve repair [116].

Similar results have been achieved through 3D printed nerve conduits made from both chitosan and synthetic polymers [117]. When chitosan was printed into the outer microporous tube and polyglycolic acid (PGA) as the inner guiding filler, the two-component artificial conduit could bridge and repair a 30 mm sciatic nerve defect in dogs with nerve continuity recovery and target muscle re-neurotization [117,118]. For example, a chitosan-PGA nerve graft could repair a 10-mm defect in rat sciatic nerve defect and maintain the continuity of the nerve tissue for 3 to 6 months. By the combination of neuronal supportive cells with the 3D bioprinted chitosan-containing constructs, the grafts could repair even larger nerve defects. A chitosan/polylactic acid-glycolic acid (PLGA)-based graft together with autologous bone marrow mesenchymal cells could repair 50 mm long sciatic nerve gaps in dogs [119]. Six months after implantation, the damaged nerves and motor functions were restored with innervated target muscles. The graft could effectively bridge a 60-mm sciatic nerve defect in dogs with the similar repair results to those of autotransplants [120]. 

Furthermore, transplantation of Schwann cells derived from dorsal root ganglion into a nerve repair graft constructed by PLGA/chitosan could increase the diameter and behavior area of axons and improve motor function after sciatic nerve injury in rats [121]. The degradation products of chitosan, i.e., chitosan oligosaccharide, or chitooligosaccharide, had protective effects against neurotoxicity. These degradation products could protect hippocampal neurons from apoptosis [122,123]. Additionally, chitooligosaccharides could increase the survival and proliferation of Schwann cells, enhance the formation of myelin in axons, and accelerate the release of neurotrophic factors, such as brain-derived neurotrophic factors and nerve growth factors [124]. Direct intravenous injection of chitooligosaccharide after common peroneal nerve injury in rabbits could activate the muscle tissues, and enhance the number of myelinated nerve fibers, as well as the thickness of myelin sheath. The cross-sectional area of tibial posterior muscle fibers was obviously augmented [125,126,127]. A chitooligosaccharide filled silica gel tube could repair a 10-mm sciatic nerve defect in rats and promote peripheral nerve regeneration. It is supposed that the beneficial effect of chitooligosaccharides is to establish an allowable microenvironment by stimulating the proliferation of Schwann cells, and increasing macrophage infiltration [128,129,130,131].

### 4.6. Hepatic Tissue and Organ Restoration 

The liver is an important biochemical reactor in the human body. It is a complex inner organ with more than six cell types and abundant vascular and biliary networks, which makes the liver regeneration extremely difficult. Liver injury is harmful and sometimes fatal. At present, the worldwide seriously scanty of orthotopic liver donors has exacerbated the demand for new therapeutic treatment for acute and chronic liver failures [132]. Because the formation of thrombus can lead to obstruction and decrease the efficiency of blood transportation, the design of hierarchical vascular networks and anti-thrombotic extracellular components are essential aspects for liver 3D bioprinting. 

In nature, hepatocytes exist in a complicated environment, surrounded with different types of ECMs. Hepatocytes are anchor-dependent cells, especially sensitive to surrounding environments for preserving their viability and proliferability [133,134,135]. Chitosan, as a kind of natural biomaterial with specific properties, has broad applications in liver tissue and organ construction and restoration. In some earlier studies, chitosan/collagen matrix generated in the *N*-hydroxysuccinimide (NHS) buffer system using cross-linking agent 1-ethyl-3-(3-dimethylaminopropyl) carbodiimide (EDC) had the potential for liver tissue regeneration [37]. The EDC cross-linked chitosan/collagen matrix presented moderate mechanical properties, excellent biocompatibility with hepatocytes. Chitosan-collagen-heparin matrix demonstrated a superior vascular biocompatibility for liver tissue construction. Chitosan modified with galactose residues could improve the adhesion of hepatocytes and maintain the viability of hepatocytes. Park et al. demonstrated that the specific interaction between asialoglycoprotein receptor (ASGPR) and galactose ligand of glycol chitosan (GC), led to the synthesis of new ECM for hepatocyte adhesion [136]. Chung et al. proposed the potential to improve the short-term viability of hepatocytes cultured on alginate/chitosan scaffolds [137]. Kim et al. reported a long-term enhancement of hepatocyte function in alginate/chitosan scaffolds [132]. Yu et al. 3D printed scaffold-free “tissue strands” as a “bioink” [138]. Co-culturing of hepatocytes with fibroblasts on an alginate/chitosan scaffold could boost the sphere formation speed. Li et al. reported that fructose coupled to porous chitosan scaffolds through the reaction of amino groups and aldehyde groups could improve the hepatic functions [139]. Fructose is a specific ligand of ASGPR in hepatocytes. The fructose-modified chitosan could induce hepatocyte aggregation, and specific metabolic activity of the artificial liver tissues to some degree. Especially, hepatocytes on ammonia-treated chitosan/collagen membranes demonstrated polar growth capabilities (Figure 8).

Over the past decade, chitosan has become one of the main components of “bioinks” for tissue and organ 3D bioprinting because of its similar biochemical properties to glycosaminoglycans (GAGs) in ECMs (Table 1) [140,141,142,143,144,145,146,147,148,149,150,151,152,153,154,155,156,157,158,159,160]. Liver 3D bioprinting is a manufacturing process with the ultimate goal to treat injured or failed livers. The main objective of liver 3D bioprinting is to develop biological liver substitutes or analogues in which patient plasma can circulate inside the vascularized hepatic tissues with metabolically active hepatocytes. An important aspect of 3D liver bioprinting is to select right cell types, such as primary hepatocytes, vascular stem cells, and hepatic stem cells. The patient-derived primary hepatocytes are a useful cell source for bioartificial liver 3D bioprinting. Many researchers have tried to optimize the cell survival environments to maintain the biochemical functions of the primary hepatocytes, so that they can carry out as many physiological functions as possible [133,134]. For example, compared with pure chitosan/collagen and chitosan/gelatin hydrogels, the chitosan/collagen/heparin and chitosan/gelatin/heparin hydrogels containing vascular cells, such as endothelial cells and smooth muscle cells, may have more potentials for liver 3D bioprinting with hierarchical vascular network construction [8,37,38]. The micro-structures of the chitosan/collagen/heparin and chitosan/gelatin/heparin hydrogels could provide anticoagulant functions before the inner surface of the vascular network is fully endothelialized. Basically, both the pure chitosan/collagen, chitosan/gelatin and chitosan/collagen/heparin, chitosan/gelatin/heparin hydrogels have large surface-volume ratio for hepatocytes to survive, since hepatocytes tend to adhere to specific substrate to migrate and proliferate. The micro-structures of the chitosan-containing molecules are beneficial to the transportation of nutrition and oxygen. 

Chitosans, as additives of gelatin-based hydrogels have been frequently used as cell-laden “bioinks” for liver tissue and organ 3D bioprinting because of their good biocompatibility (i.e., no-toxicity, non-immunogenicity), chemical gelling capability (i.e., crosslinkability), moderate biodegradability, and ECM component simulation property. The first chitosan application in liver tissue 3D bioprinting is in 2003 (Figure 9) [154]. The obstacle for chitosan solutions to be printed alone is that their physical sol-gel transitions are too low to be below 0 °C, and it is difficult for the chitosan solutions to be printed with cells at room temperatures [143,144,145]. Physical blending of chitosan and gelatin solutions is necessary, endowing the composite gelatin/chitosan hydrogel with a higher phase transition temperature (i.e., sol-gel transition point ≈ 28 °C). The biophysical and biochemical characters of chitosan molecules are vital for generating chitosan-based polyelectrolytes for the layer-by-layer 3D deposition techniques. 

Since 2005, various chitosan-based composite “bioinks,” such as chitosan/gelatin, chitosan/gelatin/alginate chitosan/gelatin/alginate/dextron-40 (glaycerol or dimethyl sulfoxide), have been explored in our laboratory through several extrusion-based 3D bioprinters (Figure 10) [155]. During the 3D bioprinting processes, the viscosity of the chitosan-based hydrogels depends largely on the composite polymer concentration, molecular weight, and cell density. These directly affect the 3D printing accuracy of the cell-laden constructs. After 3D bioprinting, the 3D printed constructs need to be stabilized using physical, chemical, and/or enzymic methods. The physical method includes thermosensitive transformation of the polymer molecules. While chemical and enzymic methods employ crosslinking agents or enzymes to crosslink or polymerize the incorporated polymer molecules. include chemical and enzymatic crosslinking. 

Most of the stabilization processes contain two steps, both the thermosensitive physical and ionic chemical crosslinks. For example, the chitosan molecules can be chemically crosslinked using tripolyphosphate (TPP), oxidized dextran or other oxidized carbohydrates, 1,1,3,3-tetramethoxypropan, and genipin after a physical sol-gel transformation, ensuring the cell-laden constructs to be stable enough for being cultured in a liquid medium [42,43]. Further blending of chitosan with other natural polymers, such as fibrinogen, hyaluronan, endows the incorporated cells with more specific physiological functions [146,147,148,149,150,151,152,153,154,155]. 

Especially, the chemical crosslinking procedures for the 3D printed constructs can be changed depending on the composite polymer components. For example, a cell-laden chitosan/gelatin/alginate construct can be crosslinked by both TPP and CaCl_2_. While, the cell-laden chitosan/gelatin/alginate/fibrinogen construct can be crosslinked by TPP, CaCl_2_, and thrombin, respectively or in combination. During the chemical crosslinking procedures, the chitosan molecules are crosslinked by TPP. The alginate molecules are crosslinked by CaCl_2_. Meanwhile the fibrinogen molecules are crosslinked by thrombin. The more crosslinks, the more stable of the 3D printed constructs. Ten years later, these classical “bioink” formulations and crosslinking strategies have been widely adapted by many other groups all over the world [146,147,148,149,150,151,152,153,154,155]. 

For complex liver 3D bioprinting, multiple networks, including vascular, neural, and biliary, should be enclosed (Figure 11) [160]. This is vital for this bionic liver to be connected to the host tissues with anti-suture and anti-stress properties. The chitosan containing ECM similar hydrogels with a large amount of water are vital for hepatocytes to anchor and survive [154,155]. Within these hydrogels, the 3D printed hepatocytes have the similar physiological functions, such as albumin, proteoglycan, and fibronectin, secretion. Meanwhile, the implantable bioartificial livers made from 3D bioprinting demonstrate the potential of permanent liver replacement and restoration. 

## 5. Challenges and Perspectives

Like building “a nuclear plant,” organ 3D bioprinting needs to face the selection and assembling of multiple polymeric materials, heterogeneous cell types, and other bioactive agents. Although significant progress has been made in vascularized and innervated organ 3D bioprinting over the last decade, some challenges remain in biomimicking each of the complex organs of human beings with special architectures, multi-cellular components, and physiological functions. The main challenges that remain in complex organ 3D bioprinting can be classified into the following three aspects: (1) Immunological rejections (or immune reactions) and other side effects from the 3D bioprinted biomaterials, especially living cells and polymeric hydrogels; (2) powerful 3D “bioprinters” that can recapitulate all the critical factors in a complex organ, such as the hierarchical biliary networks in the liver, the delicate lymphatic networks in the skin, and the multi-functional tubules in the kidney; (3) long-term growth and restoration capabilities of the bioartificial organs.

For patient with a failure organ, there are limited supply of autologous adult organ cells. Extensive donor site morbidity and complication may arise from heterogeneic cell transplantation and non-biocompatible polymer implantation. Cell viability in the 3D printed constructs directly affects the final organ growth and restoration results. The combination of chitosan-based polymers with autologous stem cells, growth factors, and other bioactive agents as predefined “bioinks” is an effective way to solve the risk of immune reactions, and teratoma formations in the bioartificial organs. Long-term stability of the 3D constructs needs to be further certified [161,162,163,164,165,166]. 

Beside the cell and polymer selection, the currently available 3D bioprinters are obviously cannot provide all the necessary capabilities for multiple cell type, polymeric material, and geometrical structure integration as those in a natural complex organ. It is challenging to develop new powerful “bioprinters” to print more cell types or stem cells/growth factors along with the selected polymeric materials for each solid organ construction with a full spectrum of physiological functions [167,168,169,170]. 

In the future, organ 3D bioprinting will play an essential role in many pertinent sciences and technologies, such as adult and stem cell biology, tissue science and engineering, drug screening and delivery, energy metabolism and detection. Especially, various sophisticated 3D bioprinters will be developed to recapitulate the macro- and micro-environments of natural organs. The prominent features of the sophisticated 3D bioprinters are the capabilities to integrate more structural, material, and physiological functions in an organic construct, and to enhance the cell loading and organizing efficiencies for multiple tissue formation, maturation, and coordination in a predefined construct [12,13,14,128,129,130,131,151,152,153,154,155].

In the future, the 3D printed bioartificial organs will be used for a plenty of clinical purposes, such as failure organ restoration and cancer defect repair. One of the major advantages of the 3D printed bioartificial organs is that cells can be extracted from the patients themselves. Stem cells derived from patient themselves will be ideal cell sources. Immune rejection raised by the recipient’s immune system can be excluded. By using the medical image, it is possible to reproduce the actual structure of the native organ of the patient. 

In the future, the 3D printed bioartificial organs will offer a great potential for clinical applications, such as high throughput drug testing and screening, disease mechanism analysis and diagnosis, surgical intervention guidance and judgement [171,172,173,174,175]. Doctors and surgeons can practice and conduct experiments with these bioartificial organs to mimic the organ transplantation procedures. Scientists can test and use new drugs with these bioartificial organs to improve the therapeutic effects and eliminate the misdiagnosis complications.

In the future, there will be standardized global database for organ bank or library which allows access to all relevant medical images. Reverse organ manufacturing may become a hot issue for those intrinsic or accidental organ failures. Customize CAD models can either derive directly from the outcomes of a symmetrical organ or portrait. By using the advanced 3D bioprinters, the average life span of human beings will be significantly prolonged. The combination of architectural predesign, material optimization, medical imaging, advanced 3D bioprinting, and robotic surgery therefore will be a popular solution to most medical problems and should be a major research field in the forthcoming scientific and technological areas.

## 6. Conclusions

Chitosan, as a resourced natural molecule has been widely explored for biomedical applications. The prominent characteristics of chitosan such as non-toxic, biodegradable, antibiosis, and structural similarities to ECMs, make it a favorable candidate for a variety of tissue repair and 3D organ bioprinting applications. Especially, the structural similarities of chitosan with glycosaminoglycan are a favorable factor for tissue repair and organ 3D bioprinting. The 3D printed chitosan-containing porous scaffolds and cell-laden constructs have particular usages in promoting skin regeneration, wound healing, bone rehabilitation, cartilage reconstruction, nerve repair, and liver restoration. The biochemical and physiological properties of the chitosan-containing “bioinks” can easily be adjusted by changing the chitosan molecules in the hydrogels. It is expected that further study of chitosan molecules and their association with other polymers will reveal greater prospects of this unique polymer in complex organ manufacturing and clinical applications. 

## Figures and Tables

**Figure 1 micromachines-10-00765-f001:**
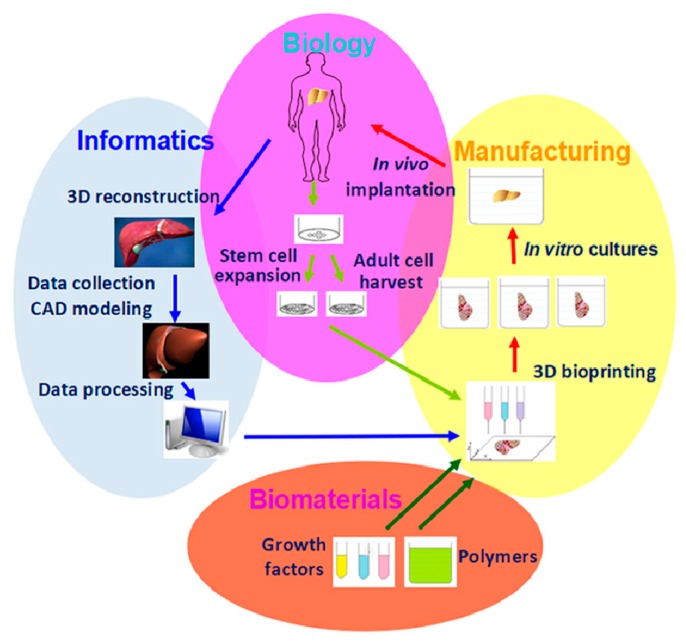
Graphical description of organ 3D bioprinting with four major aspects: cell extraction (i.e., biological), data collection (i.e., informatical), starting material preparation (i.e., biomaterial), and manufacturing (i.e., processing) parts.

**Figure 2 micromachines-10-00765-f002:**
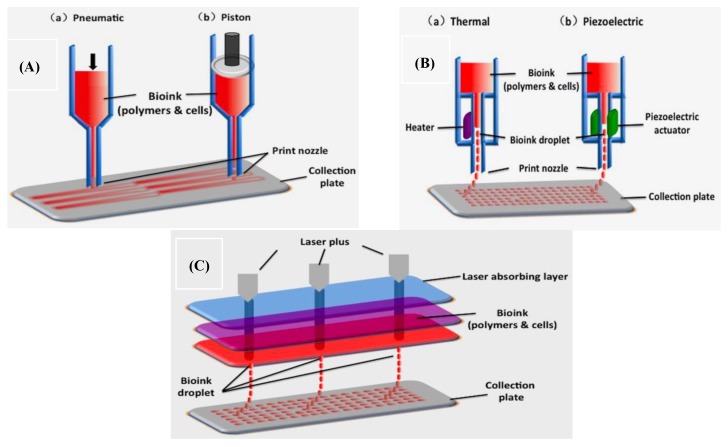
A schematic diagram of the three major types of organ 3D bioprinting technologies: (**A**) two-nozzle extrusion-based bioprinting (a: pneumatic; b: piston); (**B**) two-nozzle inkjet-based bioprinting (a: heater; b: piezoelectric actuator); (**C**) three-channel laser-assisted bioprinting. Image reproduced with permission from [26].

**Figure 3 micromachines-10-00765-f003:**
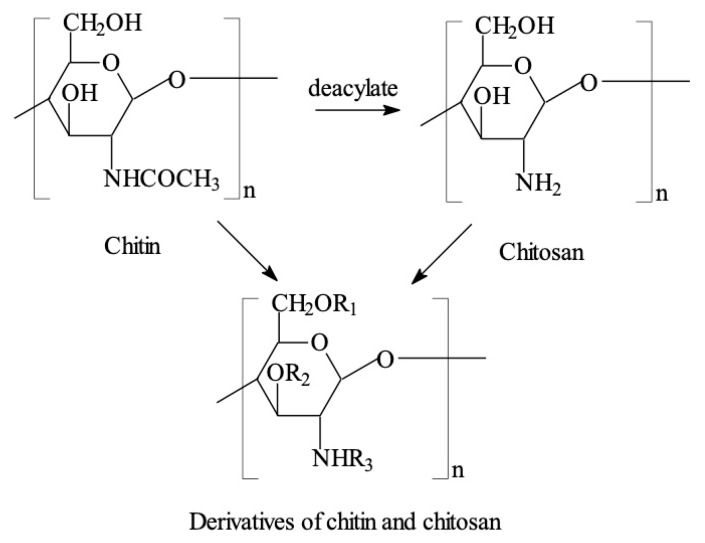
Chitosan generated by deacylation of chitin can be chemically modified through changing the functional groups, R_1_, R_2_, R_3_ in the derivatives represents -COCH_3_, -CH_3_, -CH_2_COOH, -SO_3_H, -PO(OH)_2_, etc. Image reproduced with permission from [31].

**Figure 4 micromachines-10-00765-f004:**
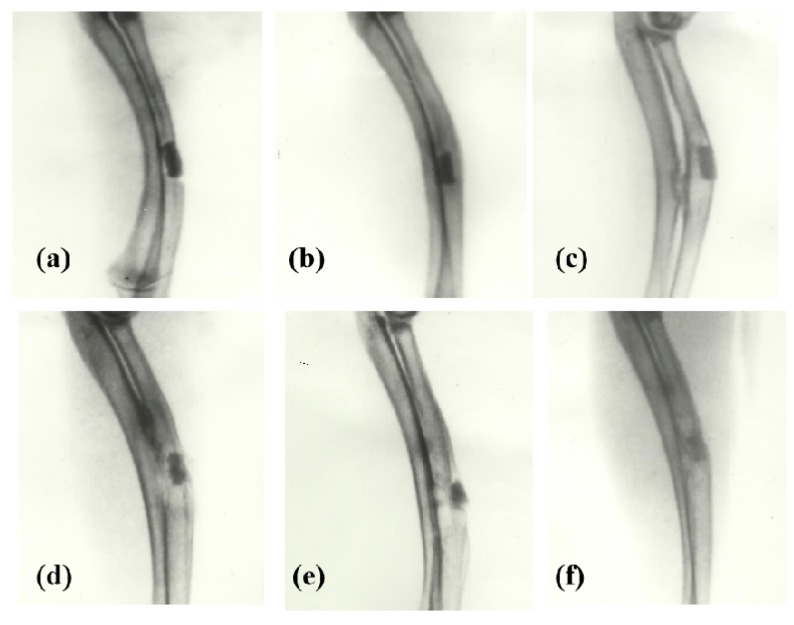
X-ray radiographs of large bone repair in rabbits with phosphorylated chitosan (P-chitosan) containing biodegradable calcium phosphate cements (CPCs): (**a**) 1 week (0.12 g/mL P-chitosan); (**b**) 4 weeks (0.14 g/mL P-chitosan); (**c**) 12 weeks (0.12 g/mL P-chitosan); (**d**) 12 weeks (0.07 g/mL P-chitosan); (**e**) 12 weeks (0.02 g/mL P-chitosan); (**f**) 22 weeks (0.12 g/mL P-chitosan). Image reproduced with permission from [33].

**Figure 5 micromachines-10-00765-f005:**
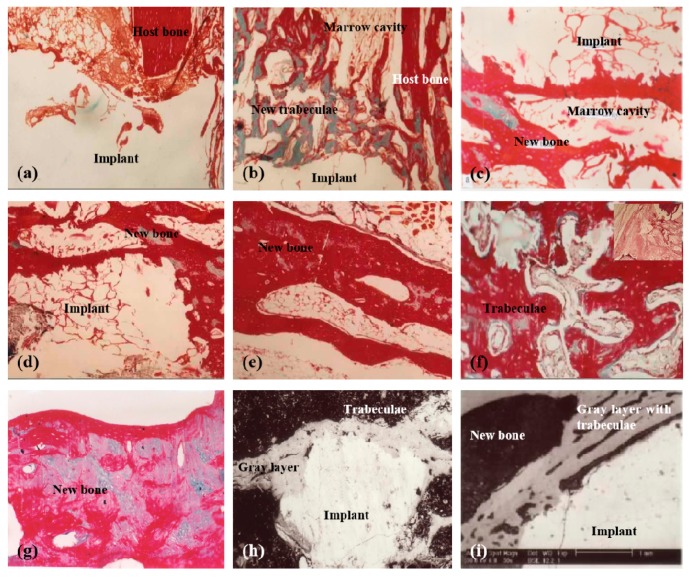
Tissue responses to different samples containing different concentrations of phosphorylated chitosan (P-chitosan) at different time points after implantation: (**a**) 1 week (0.12 g/mL P-chitosan), Masson Trichroism (M-T) staining; (**b**) 4 weeks (0.12 g/mL P-chitosan), M-T staining; (**c**–**e**) 12 weeks (0.02 g/mL P-chitosan); (**f**) 12 weeks (0.12 g/mL and 0.05 g/mL P-chitosan respectively), M-T and haematoxylin-eosin staining; (**g**) 22 weeks (0.12 g/mL P-chitosan) M-T staining; (**h**) 12 weeks (0.02 g/mL P-chitosan), a back scattered scanning electron microscopy (BSE) image; (**i**) 12 weeks (0.02 g/mL P-chitosan), a BSE image. Image reproduced with permission from [33].

**Figure 6 micromachines-10-00765-f006:**
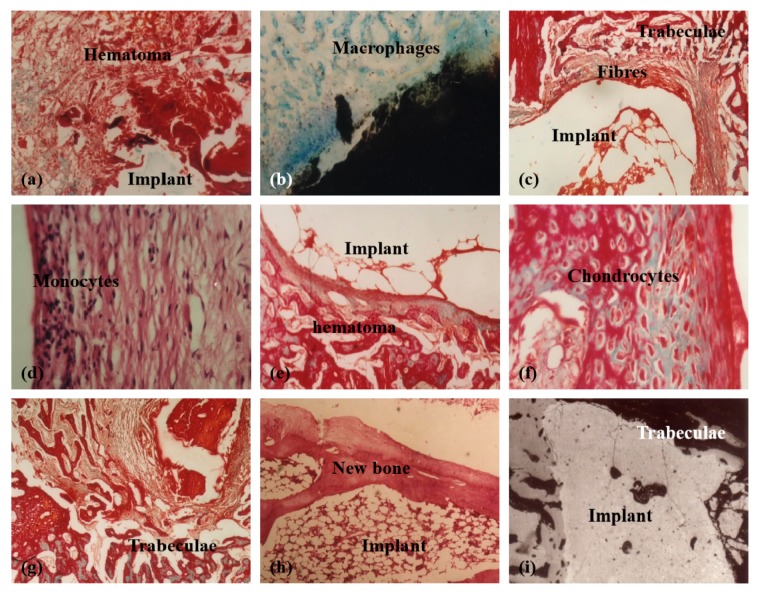
Tissue responses to different samples containing different concentrations of phosphorylated chitin (P-chitin) at different time points after implantation: (**a**,**b**) 1 week (0.14 g/mL P-chitin), Masson Trichroism (M-T) and Giemsa staining respectively; (**c**,**d**) 4 weeks (0.14 g/mL P-chitin), M-T and haematoxylin-eosin staining respectively; (**e**,**f**) 4 weeks (0.08 g/mL P-chitin); (**g**) 12 weeks (0.08 g/mL P-chitin) M-T staining; (**h**) 22 weeks (0.14 g/mL P-chitin); (**i**) 12 weeks (0.02 g/mL P-chitin), a back scattered scanning electron microscopy image. Image reproduced with permission from [32].

**Figure 7 micromachines-10-00765-f007:**
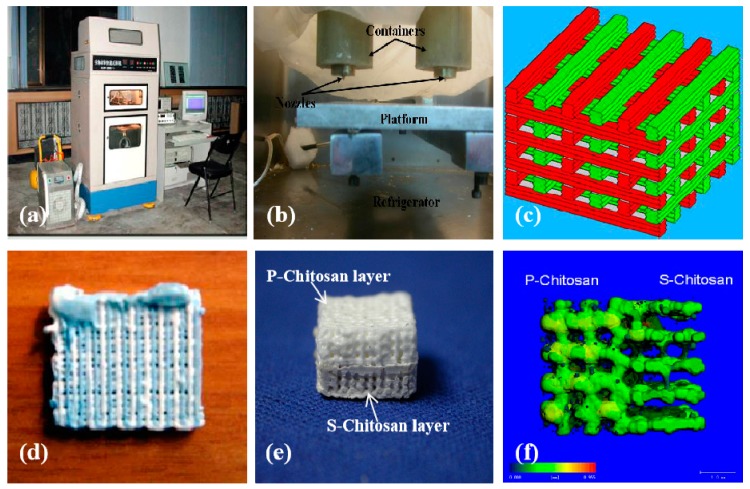
Graphical description of large bone repair scaffolds made in Tsinghua University, the corresponding author’s laboratory in 2007: (**a**) a double-nozzle low-temperature 3D bioprinter; (**b**) working state of the double nozzles; (**c**) a grid computer-aided design (CAD) model containing two material systems; (**d**) a sample made from chitosan/gelatin and polyurethane; (**e**) a sample containing both P-chitosan and S-chitosan made via the double-nozzle low-temperature 3D bioprinter; (**f**) a computerized tomography of the 3D printed sample of (**e**). Image reproduced with permission from [87].

**Figure 8 micromachines-10-00765-f008:**
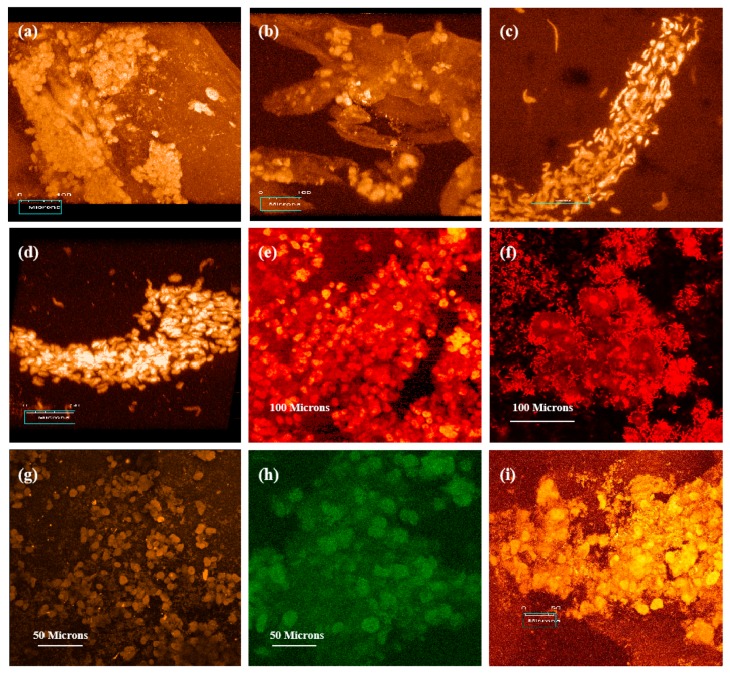
Laser scanning confocal microscope observations (propidium iodide staining) of hepatocytes seeded on the matrices of (**a**) ammonia treated collagen, (**b**) heparin sodium containing ammonia treated collagen, (**c**,**d**) ammonia treated chitosan/collagen, (**e**) heparin sodium containing ammonia treated chitosan/collagen, (**f**) sodium hyaluronate containing ammonia treated collagen/chitosan, (**g**,**h**,**i)** ammonia treated chitosan after 25 days of culture. Image reproduced with permission from [38].

**Figure 9 micromachines-10-00765-f009:**
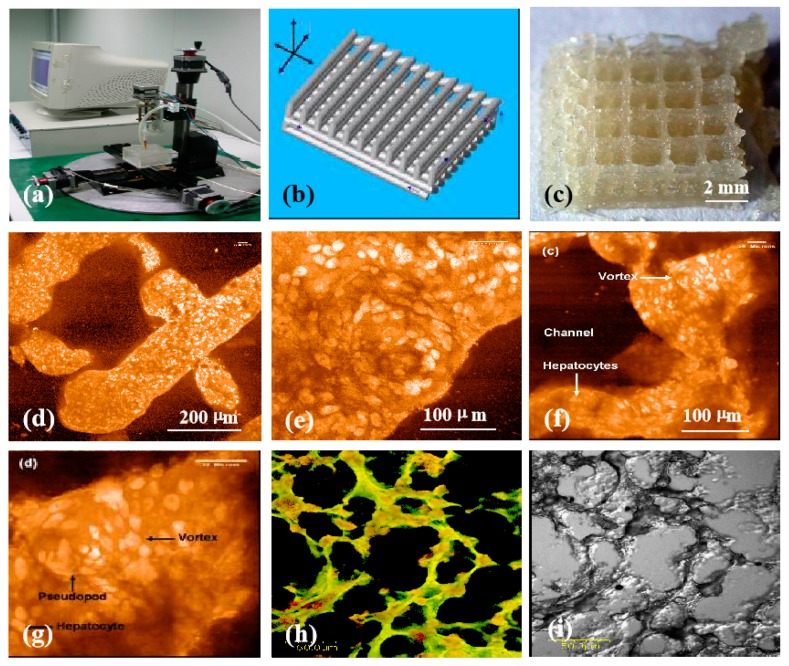
Three-dimensional (3D) printing of hepatocyte-laden chitosan/gelatin hydrogels and laser scanning confocal microscope observations of hepatocytes in the 3D printed chitosan/gelatin constructs with propidium iodide staining: (**a**) The 3D bioprinter made in the corresponding author’s laboratory in 2003; (**b**) a grid computer-aided design (CAD) model; (**c**) a grid cell-laden 3D construct immediately after 3D bioprinting; (**d**,**e**) hepatocytes in the 3D printed chitosan/gelatin construct 1 month after in vitro culture; (**f**–**i**) hepatocytes in the 3D printed chitosan/gelatin construct 2 months after in vitro culture; (**e**,**g**) are the magnifications of (**d**) and (**f**) respectively; (i) a dark-field micrograph of (**h**). Image reproduced with permission from [154].

**Figure 10 micromachines-10-00765-f010:**
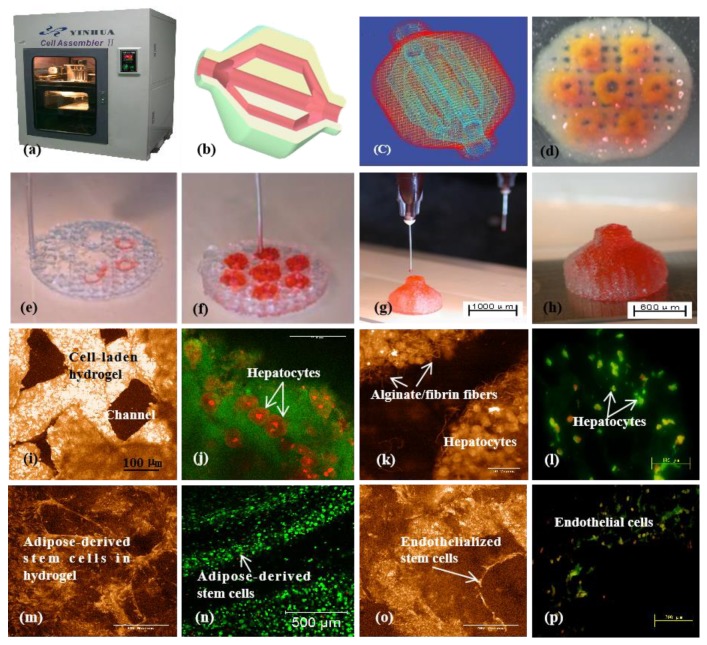
A large scale-up 3D printed vascularized liver tissue constructed through the double-nozzle 3D bioprinter created in Tsinghua University, the corresponding author’s laboratory in 2006: (**a**) the double-nozzle extrusion-based 3D bioprinter; (**b**) a computer-aided design (CAD) model containing a hierarchical vascular network; (**c**) a CAD model containing the branched vascular network; (**d**) a few layers of the 3D bioprinted construct containing both adipose-derived stem cells (ASCs) encapsulated a gelatin/alginate/fibrin hydrogel and hepatocytes encapsulated in a gelatin/alginate/chitosan hydrogel; (**e**–**h**) 3D printing process of a semi-elliptical construct containing both ASCs and hepatocytes encapsulated in different hydrogels (i.e., gelatin/alginate/fibrin and gelatin/alginate/chitosan); (**i**–**l**) hepatocytes encapsulated in the gelatin/alginate/chitosan hydrogel after 3D bioprinting and different periods of in vitro cultures; (**m**–**p**) ASCs encapsulated in the gelatin/alginate/fibrin hydrogel after 3D bioprinting and different periods of in vitro cultures as well as growth factor inductions. Image reproduced with permission from [155].

**Figure 11 micromachines-10-00765-f011:**
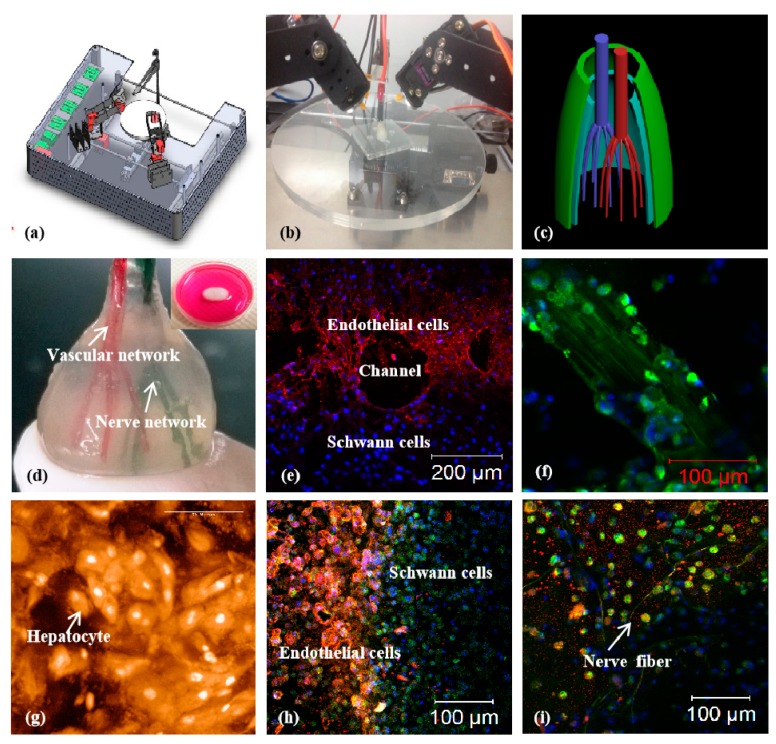
A combined four-nozzle organ three-dimensional (3D) bioprinting technology created in Tsinghua University, the corresponding author’s laboratory in 2013 [160]: (**a**) equipment of the combined four-nozzle organ 3D bioprinter; (**b**) working state of the combined four-nozzle organ 3D printer; (**c**) a computer-aided design (CAD) model representing a large scale-up vascularized and innervated hepatic tissue; (**d**) a semi-ellipse 3D construct containing a poly (lactic acid-co-glycolic acid) (PLGA) overcoat, a hepatic tissue made from hepatocytes encapsulated in the gelatin/chitosan hydrogel, a branched vascular network with a fully confluent endothelialized adipose-derived stem cells (ASCs) on the inner surface of the cell-laden gelatin/alginate/fibrin hydrogel and a hierarchical never network made from Shwann cells encapsulated in the gelatin/hyaluronate hydrogel, the maximal diameter of the semi-ellipse construct can be adjusted from 1 mm to 2 cm according to the CAD model; (**e**) a cross section of (**d**), showing the interface of endothelialized ASCs and Schwann cells around a branched channel; (**f**) a large bundle of nerve fibers formed in the hierarchical never network of (**d**); (**g**) hepatocytes within the gelatin/chitosan hydrogel and underneath the PLGA overcoat; (**h**) a magnified picture of (**d**) showing the interface between the endothelialized ASCs and Schwann cells; (**i**) some thin nerve fibers near the hepatocyte-laden gelatin/chitosan hydrogel.

**Table 1 micromachines-10-00765-t001:** Chitosan containing “bioinks” for organ 3D bioprinting.

“Bioink” Formulation	3D Bioprinting Technique	Cross-Linking Method	Application	Ref
Chitosan/gelatin/hepatocytes	One nozzle extrusion-based 3D bioprinting	3% sodium tripolyphosphate (TPP)	Large scale-up hepatic tissue manufacturing	[154]
Chitosan/gelatin/alginate/hepatocytes and gelatin/alginate/fibrinogen/adipose-derived stem cells (ASCs)	Two nozzle extrusion-based 3D bioprinting	Triple crosslinking with TPP/CaCl_2_/thrombin solutions after 3D bioprinting	Vascularized hepatic tissues with hierarchical branched vascular networks	[155]
Chitosan/sodium alginate (CS-SA) hydrogels	Rapid Prototyping (Fab@Home) printer	10% CaCl_2_ (*w/v*) solution	Skin tissue regeneration	[156]
Oxidized hyaluronate (OHA)/glycol chitosan (GC)/adipic acid dihydrazide (ADH) hydrogels	3D bioprinter (Invivo^®^, Rokit, Korea)	Not require any post-gelation or additional cross-linking	Self-healing hydrogel system for cartilage regeneration	[157]
Chitosan-hydroxyapatite hydrogels (Chitosan-HA)	The Fab@Home™ (The Seraph Robotics, USA) open source RP platform Model 3	2% CaCl_2_ (*v/v*) for 15 min	Bone regeneration	[158]
Chitosan/alginate hydrogel	One/two nozzle extrusion-based 3D bioprinting	CaCl_2_ solution	Vessel-like tubular microfluidic channels	[159]
Polylactic acid-co-glycolic acid (PLGA)-gelatin/alginate/fibrinogen/ASCs-gelatin/chitosan/hepatocytes-gelatin/hyaluronate/Schwann cells	Combined four-nozzle 3D bioprinting	Triple crosslinking with TPP/CaCl_2_/thrombin solutions after 3D bioprinting	Vascularized and innervated liver tissue generating	[160]
Chitosan film enhanced chitosan nerve guides (CFeCNGs)	REAXON^®^ Nerve Guide	Not required	Nerve regeneration	[115]

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
