# Peer review of "Chitosans for Tissue Repair and Organ Three-Dimensional (3D) Bioprinting"

_micromachines, 2019, doi:10.3390/mi10110765_

Round 1

Reviewer 1 Report

The authors reviewed the applications of the biomacromolecule Chitosan in tissue engineering and organ reconstruction by 3D bioprinting techniques. The review is comprehensive, and the topic is important, so I recommend its publications in Micromachines after the following comments are addressed:

Chitosan is one of the most commonly used natural polymers, could the authors give a brief description of other natural polymers as a background for better comparison and understanding?

The font size of the text in Figure 2 is too small to read.

The English expression should be double checked, for example, line 100 “Due to the specific properties of natural polymers, such as biocompatible, biodegradable, bioavailable and biostable, natural polymers, such as alginate, gelatin, hyaluronate, and chitosan, have been chosen as the preferable candidates for organ 3D bioprinting” should be rephrased.

In Figure 4, there is very limited information can be obtained. The authors should add more captions to illustrate the content.

Is it possible to give more specific solutions in the perspective part? As the challenges are well-known but the future directions are not so clear in detail.

Author Response

Comment: The authors reviewed the applications of the biomacromolecule Chitosan in tissue engineering and organ reconstruction by 3D bioprinting techniques. The review is comprehensive, and the topic is important, so I recommend its publications in Micromachines after the following comments are addressed:

Response: Thanks a lot for the faithful comments.

Comments: Chitosan is one of the most commonly used natural polymers, could the authors give a brief description of other natural polymers as a background for better comparison and understanding?

Response: A brief description of other natural polymers as a background for better comparison and understanding has been added in the Introduction (Page 2, line 63-75).

Comments: The font size of the text in Figure 2 is too small to read.

Response: Figure 2 has been reorganized to make the content clear enough (page 4, line 126-132).  

Comments: The English expression should be double checked, for example, line 100 “Due to the specific properties of natural polymers, such as biocompatible, biodegradable, bioavailable and biostable, natural polymers, such as alginate, gelatin, hyaluronate, and chitosan, have been chosen as the preferable candidates for organ 3D bioprinting” should be rephrased.

Response: The English expression has been improved throughout the paper in red color (e.g. page 3 line 101-110).

Comment: In Figure 4, there is very limited information can be obtained. The authors should add more captions to illustrate the content. 

Response: The related figure legents have been improved (Figure 4-11).

Comment: Is it possible to give more specific solutions in the perspective part? As the challenges are well-known but the future directions are not so clear in detail.

Response: More specific solutions in the perspective part have been presented (page 21-22). 

Reviewer 2 Report

The paper is a review of Chitosans developed through 3D printing and used for tissue and organ repair.  The review is extensive and covers the area well, however their are a number of suggestions and revisions the authors should make to improve the paper.

Firstly, the last sentence in the abstract does not necessarily reflect the paper's contents as it focuses primarily on applications of Chitosans rather than the properties and processes used for 3D printing.

The introduction of the paper makes no mention of 3D printing or chitosans, therefore poorly motivating the work.  There is also no outline for what the reader may expect in the paper that should be included in the introduction.

Many figures are not well explained in the text, for instance Figure 1 is not explained.

At the beginning of section 2, not all 3D printers build structures in strictly a layer-by-layer fashion or convert into two dimensional planes prior to printing, as suggested by the paper. 

In section 2.2 the authors state a number of challenges and bottleneck problems, not all of these have been overcome with 3D printing completely, as suggested by the paper.

Prior to Section 3, the authors have poor motivation for why 3D printed chitosan is of importance to the reader.

In section 4.2 the authors state 3D bioprinting chitosan accelerates the wound healing process, what is this acceleration in comparison to?

In several areas the paper would benefit with more quantitative evidence provided, for instance in Section 4.3.3 the authors state the CSNF/PPC scaffolds performs significantly higher than those with PPC, but provides no quantification for the reader to understand this claim.  Additionally authors state a "large number" of osteoblasts adhered to a scaffold surface without providing proper quantification.

In section 3.3 as well, and in Figure 9 the name "Professor Wang" is highlighted which is odd and biased because researchers are not mentioned by name otherwise in the paper.  This name mention should be removed or extensive further credit needs to be highlighted by name from other contributors in the field not linked to authors on this paper.

In the challenges and perspectives section the authors mention challenges for single or double-nozzle 3D bioprinters, but little of the 3D printing challenges are otherwise talked about in the manuscript.  Including more about these processes to support the extensive focus on applications would significantly improve the manuscript.

Author Response

Comments: The paper is a review of Chitosans developed through 3D printing and used for tissue and organ repair. The review is extensive and covers the area well, however their are a number of suggestions and revisions the authors should make to improve the paper.

Response: The comments are reasonable and the quality of the paper has been improved.

Comments: Firstly, the last sentence in the abstract does not necessarily reflect the paper's contents as it focuses primarily on applications of Chitosans rather than the properties and processes used for 3D printing. 

Response: The related part has been revised (page 1, line 19-23).

Comments: The introduction of the paper makes no mention of 3D printing or chitosans, therefore poorly motivating the work.  There is also no outline for what the reader may expect in the paper that should be included in the introduction.

Response: The related parts have been added (page 2, line 57-75).

Comments: Many figures are not well explained in the text, for instance Figure 1 is not explained.

Responses: All the figures have been well explained in the text (page 2, line 86-89; page 4, line 118-124; page 5, line 157-167; etc).  

Comments: At the beginning of section 2, not all 3D printers build structures in strictly a layer-by-layer fashion or convert into two dimensional planes prior to printing, as suggested by the paper.

Response: The expression has been changed (page 2, line 81).

Comments: In section 2.2 the authors state a number of challenges and bottleneck problems, not all of these have been overcome with 3D printing completely, as suggested by the paper.

Response: The expression has been changed (page 3-4, line 114-124).

Comments: Prior to Section 3, the authors have poor motivation for why 3D printed chitosan is of importance to the reader.

Response: The expression has been revised (page 4, line 139-140).

Comments: In section 4.2 the authors state 3D bioprinting chitosan accelerates the wound healing process, what is this acceleration in comparison to?

Response: The expression has been revised (page 7, line 268-269).

Comments: In several areas the paper would benefit with more quantitative evidence provided, for instance in Section 4.3.3 the authors state the CSNF/PPC scaffolds performs significantly higher than those with PPC, but provides no quantification for the reader to understand this claim.  Additionally authors state a "large number" of osteoblasts adhered to a scaffold surface without providing proper quantification.

Response: The expression has been revised (page 8, line 309-317).

Comments: In section 3.3 as well, and in Figure 9 the name "Professor Wang" is highlighted which is odd and biased because researchers are not mentioned by name otherwise in the paper.  This name mention should be removed or extensive further credit needs to be highlighted by name from other contributors in the field not linked to authors on this paper.

Response: The names of "Professor Wang" have been deleted in the related contents.

Comments: In the challenges and perspectives section the authors mention challenges for single or double-nozzle 3D bioprinters, but little of the 3D printing challenges are otherwise talked about in the manuscript.  Including more about these processes to support the extensive focus on applications would significantly improve the manuscript.

Response: the challenges and perspectives section has been revised properly (page 22-23, line 690-717).
